# Fusion of Normoxic- and Hypoxic-Preconditioned Myoblasts Leads to Increased Hypertrophy

**DOI:** 10.3390/cells11061059

**Published:** 2022-03-21

**Authors:** Tamara Pircher, Henning Wackerhage, Elif Akova, Wolfgang Böcker, Attila Aszodi, Maximilian M. Saller

**Affiliations:** 1Department of Orthopaedics and Trauma Surgery, Musculoskeletal University Center Munich (MUM), Ludwig-Maximilians-University (LMU), Fraunhoferstraße 20, 82152 Planegg-Martinsried, Germany; tamara.pircher@med.uni-muenchen.de (T.P.); elif.akova@med.uni-muenchen.de (E.A.); wolfgang.boecker@med.uni-muenchen.de (W.B.); attila.aszodi@med.uni-muenchen.de (A.A.); 2Faculty of Sport and Health Sciences, Technical University of Munich, Georg-Brauchle-Ring 60, 80992 Munich, Germany; henning.wackerhage@tum.de

**Keywords:** C2C12, myoblasts, myotube, fusion, myogenic differentiation, hypoxia, oxygen

## Abstract

Injuries, high altitude, and endurance exercise lead to hypoxic conditions in skeletal muscle and sometimes to hypoxia-induced local tissue damage. Thus, regenerative myoblasts/satellite cells are exposed to different levels and durations of partial oxygen pressure depending on the spatial distance from the blood vessels. To date, it is unclear how hypoxia affects myoblasts proliferation, differentiation, and particularly fusion with normoxic myoblasts. To study this, we investigated how 21% and 2% oxygen affects C2C12 myoblast morphology, proliferation, and myogenic differentiation and evaluated the fusion of normoxic- or hypoxic-preconditioned C2C12 cells in 21% or 2% oxygen in vitro. Out data show that the long-term hypoxic culture condition does not affect the proliferation of C2C12 cells but leads to rounder cells and reduced myotube formation when compared with myoblasts exposed to normoxia. However, when normoxic- and hypoxic-preconditioned myoblasts were differentiated together, the resultant myotubes were significantly larger than the control myotubes. Whole transcriptome sequencing analysis revealed several novel candidate genes that are differentially regulated during the differentiation under normoxia and hypoxia in mixed culture conditions and may thus be involved in the increase in myotube size. Taken together, oxygen-dependent adaption and interaction of myoblasts may represent a novel approach for the development of innovative therapeutic targets.

## 1. Introduction

Since the great oxidation event about 2.4 billion years ago, oxygen has become essential for most forms of life. To date, most organisms not only utilize oxygen for ATP resynthesis through oxidative phosphorylation but also sense oxygen levels and respond to hypoxia with adaptations mediated by signal transduction proteins such as hypoxia-induced factors.

In muscle tissues, and specifically in skeletal muscles, oxygen regulates functions such as proliferation, differentiation, and metabolic activity. Physiologically, hypoxia in skeletal muscle cells occurs during embryogenesis or muscle regeneration after injury. Moreover, satellite cells are located in hypoxic niches between the sarcolemma and the basal membrane, which have been shown to be essential for embryogenesis and injury-induced regeneration [1,2].

Other forms of physiological hypoxia are high altitude and exercise [3]. Exposure to altitude leads to hypoxic adaption and increased oxygen capacity in the blood—for example, by promoting erythropoietin (EPO) expression [4]—while hypoxic exposure of populations for thousands of years has resulted in the selection of mutants of genes such as *EPAS1* (Endothelial PAS Domain Protein 1) in Tibetans and other people [5].

Hypoxia can also be caused by pathological defects such as arteriosclerosis or muscle injury. Extreme muscle trauma is the most common injury seen in trauma centers and the morbidity is highly associated with ischemia, often resulting in permanent disability or even amputation [6,7]. After 6 h of ischemia, muscle is severely damaged, with outcomes ranging from loss of muscle function to necrosis [8]. During ischemia, first, phosphocreatine and glycogen decrease; lactate then increases; and eventually, ATP declines, leading to rigor mortis [8].

Immediately after reversible ischemic injury, satellite-cell-mediated regeneration is initiated [9,10]; the physiological function of satellite cells (SCs) relies on blood flow and oxygen [11]. To date, many studies have analyzed skeletal muscle regeneration only at fixed oxygen partial pressures. In real life, however, SCs and myoblasts are likely to be exposed to spatiotemporally different levels of oxygen depending on how close they are to functioning, oxygen-delivering blood vessels. Later, myoblasts that were exposed to different levels of oxygen will differentiate into myotubes at higher or lower oxygen partial pressures, depending on the regeneration and function of the vasculature. To date, it is not known how such variations of oxygen partial pressure affect myoblast proliferation and their differentiation into myotubes. In particular, we do not understand how mixed populations of normoxic- and hypoxic-conditioned myoblasts differentiate into myotubes at different oxygen levels.

Given this gap in our knowledge, we investigated the effects of 21% normoxia and 2% hypoxia on C2C12 myoblast proliferation and differentiation into myotubes at 21% normoxia or 2% hypoxia. We also analyzed how the mixing of normoxic- and hypoxic-conditioned myoblasts affects myogenic differentiation and transcription. Our data show that long-term hypoxia does not affect myoblast proliferation. Instead, it influences their differentiation into myotubes, with the greatest differences observed when mixed cultures differentiate. Hypoxia drives a unique gene expression program that includes high expression of *Car9* (carbonic anhydrase 9), a known target of hypoxia-related transcription factors.

## 2. Materials and Methods

### 2.1. Cell Culture Conditions

C2C12 immortalized mouse myoblasts (Sigma Aldrich, USA) were cultured up to 15 passages in growth medium (GM) composed of Dulbecco’s Modified Eagle Medium (DMEM) with GlutaMAX I (Thermo Fisher, USA), 10% fetal bovine serum (FBS, Sigma Aldrich, St. Louis, MI, USA), and 40 IU/mL penicillin/streptomycin (Biochrom, Berlin, Germany) at 37 °C with 5% CO_2_/21% O_2_ (normoxia) or 5% CO_2_/2% O_2_ (hypoxia). The medium was changed at least twice per week, if not stated otherwise, and cell confluence was kept below 50% to avoid spontaneous fusion of C2C12 cells.

### 2.2. Cell Proliferation and Morphology

To evaluate potential changes in cell proliferation due to continuous low oxygen concentration, we quantified cumulative population doublings (cumPD) and population doubling time (PDT) over 28 days. Therefore, 50,000 cells were seeded in T25 cell culture flasks and counted every 3 or 4 days, respectively. To assess oxygen-related differences in cell morphology, phase contrast images were taken 24 h after cell counting and the cell area, as well as the aspect ratio (AR), of at least 250 cells over all passages was quantified with ImageJ [12].

### 2.3. Generation of Stably Fluorescent C2C12 Cells

To analyze the interaction between hypoxic and normoxic cultured cells, we stably inserted the coding sequence for the green or red fluorescent protein (GFP or RFP) into C2C12 myoblasts. Therefore, 0.4 µL of sleeping beauty transposon plasmids (GFP: pSBbiGP; RFP: psBbiRP) [13], 0.9 µg transposase (pCMV(CAT)T7-SB100) [14], and 20 µL nucleofector solution (Amaxa™ Cell Line Nucleofactor™ Kit V, Lonza, Basel, Switzerland) were added to 500,000 cells and electroporated with the 4D-Nucleofector™ Core Unit (program B 032, Lonza, Basel, Switzerland). After nucleofection, cells were seeded in 6-well plates, selected with 2 µg/mL puromycin (Thermo Fisher, Waltham, MA, USA) for 10 days under normoxic condition and then sorted for high fluorescence (FACSAriaFusion, BD, Franklin Lakes, NJ, USA). For all further experiments, C2C12*^GFP^* were exclusively cultured at 21% O_2_, while C2C12*^RFP^* were cultured at 2% O_2_.

### 2.4. Myoblast Fusion Assay

To observe fusion of cells that were cultured at different oxygen concentrations, 6-well plates were seeded with 450,000 cells/well with only C2C12*^GFP^* or C2C12*^RFP^* or a combination of both (225,000 cells/well of C2C12*^GFP^* and C2C12*^RFP^*) and incubated in 21% or 2% O_2_. After 6 days of proliferation in GM, myogenic differentiation was induced by switching to the differentiation medium (DM) consisting of DMEM supplemented with 2% horse serum (HS, Sigma Aldrich, St. Louis, MI, USA) and 40 IU/mL penicillin/streptomycin. After 4 days of myogenic differentiation, the cells were fixed in 4% paraformaldehyde (PFA) in phosphate-buffered saline (PBS) for 15 min at room temperature, washed three times in PBS, and the newly formed myotubes were visualized by immunocytochemistry using an antibody against myosin heavy chain 1E (MYH1E, clone MF20, obtained from the Developmental Studies Hybridoma Bank developed under the auspices of the NICHD and maintained by The University of Iowa, Department of Biological Sciences, Iowa City, IA, USA 52242). Therefore, fixed cells were blocked with blocking solution (0.5% Triton X-100 and 10% horse serum in PBS) for 1 h at room temperature and incubated overnight at 4 °C with the primary antibody (1:50 in blocking solution). Afterwards, cells were washed three times with PBS at room temperature and incubated with appropriate fluorophore-conjugated secondary donkey anti-mouse antibodies (1:500 in blocking solution, Thermo Fisher, Waltham, MA, USA) for 1 h at room temperature. Finally, cells were washed twice in PBS, counterstained with 4′,6-diamidine-2′-phenylindole dihydrochloride (DAPI), washed once with PBS, and mounted with Fluoroshield (Abcam, Cambridge, UK). Large overview images covering an area of slightly over 0.75 cm^2^ were acquired with an epifluorescence microscope (AxioObserver, Zeiss, Jena, Germany).

To quantify differences in C2C12 fusion, the relative number of myotubes (MT: MT^GFP^, MT^RFP^, or MT^GFP+RFP^) were quantified with ImageJ (NIH, USA). Therefore, GFP and RFP image channels were converted to 8-bit grayscale, noise was reduced with a 2-pixel median filter, and a manually determined threshold including all fused cells was applied. Afterwards, the MYH1E image channel was also converted to 8-bit grayscale and merged with the newly created GFP and the RFP image channels using the image calculator, thereby only overlapping cells within both images present in the final composite (GFP and MYH1E-channel image; RFP and MHY1E-channel image). Finally, both composites were merged into one colored image using channel merger of ImageJ and counted manually with the ImageJ cell counter.

Absolute areas of GFP^+^, RFP^+^, and RFP^+^/GFP^+^ pixels in the MYH1E^+^ area were determined with a custom python script (Python 3.7.6) using the following libraries tools: numpy (for matrix processing), scikit-image (for standard image processing algorithm implementations), and matplotlib (for visualization).

All images were read into memory and single-pixel calculation was used to detect cells. Furthermore, the images were treated in the same way as in ImageJ, and the merged image was subtracted from the composites (GFP and MYH1E-channel image; RFP and MHY1E-channel image) to maintain the area for MT^GFP^ and MT^RFP^ without inclusion of MT^GFP+RFP^.

### 2.5. mRNA-Sequencing and Bioinformatic Analysis

To identify candidate genes that are differentially regulated, C2C12 cells were differentiated as described above and lysed with Trizol (Invitrogen, Waltham, MA, USA) after 24, 72, 96, or 144 h. RNA was isolated with Direct-zol™ (Zymo-Research, Irvine, CA, USA) and RNA integrity was validated with a Bioanalyzer (Agilent, Santa Clara, CA, USA). RNA-sequencing libraries were generated using the SENSE mRNA-Seq Library Prep Kit V2 (Lexogen, Vienna, Austria) according to the manufacturer’s protocol. Sequencing was performed on a HiSeq1500 device (Illumina, San Diego, CA, USA) with a read length of 50 bp and a sequencing depth of approximately 6 million reads per sample.

FASTQ files were demultiplexed by the sample-specific barcodes used for generation of each library. Reads were aligned to the *Mus musculus* genome (release GRCh38.99) using STAR (version 2.7.2b). Genes with less than 10 reads were filtered out in all samples, leaving 22,105 genes for further analysis. Normalization was performed through variance stabilizing transformation (vst) for Principal Component Analysis (PCA) and the most differentially expressed genes were selected for further clustering. Differential gene expression was analyzed by the DESeq2 package (version 1.28.1) in R software (version 4.0.3) with an adjusted *p*-value (*p*-adj) of <0.05 and a Log2FoldChange of ±2 cut-off for each condition and each time point. Venn diagram was created using significantly different expressed genes with a *p*-adj value of <0.05. Gene set enrichment analysis (GSEA) was performed at each time point through the Molecular Signatures Database (MSigDB) R package (version 7.0) using org.Mm.eg.db (version 3.11.4) and mouse_H_v5 hallmark gene set.

### 2.6. Statistical Analysis

Morphology and proliferation experiments were carried out three times in triplicates and C2C12 fusion experiments were carried out four times in triplicates. Statistical significance was calculated after determination of a Gaussian distribution using either a one-way ANOVA or a *t*-test with appropriate posthoc tests in R (version 4.1.0). Statistical significance was assumed at a *p*-value of ≤0.05. Data were represented as either the mean and the standard deviation (SD) or the median with quartiles.

## 3. Results

### 3.1. Prolonged Hypoxic Exposure Affects the Morphology of C2C12 Myoblasts but Does Not Change Their Proliferation

To evaluate whether long-term hypoxia affects cell morphology, C2C12 myoblasts were exposed to 2% oxygen for 4 weeks. Prolonged hypoxia altered the morphology of C2C12 myoblasts, as shown in Figure 1B, compared with C2C12 in normoxia (Figure 1A). Hypoxic cells had the same area (Figure 1A’,B’,C), but their aspect ratio was 16% smaller (*p* ≤ 0.05) compared to myoblasts cultured under normoxia (Figure 1D).

To determine the effect of long-term 2% hypoxia on proliferation, we quantified the cumulative population doubling and the population doubling time of C2C12 myoblasts over 28 days. Although the cumulative population doubling of hypoxic cultured C2C12 was 6.5% lower compared with normoxic cells (Figure 1E), the mean population doubling time was not significantly different (Figure 1F), suggesting that long-term hypoxia does not affect proliferation.

### 3.2. Long-Term Hypoxia Inhibits Myoblast Fusion In Vitro

Next, we investigated whether long-term hypoxia affects myogenic differentiation. We found that 2% hypoxia inhibited the formation of myotubes, when compared with normoxic myoblasts (Figure 2A). Specifically, the area of myotubes was 71.1% smaller (*p* ≤ 0.001; Figure 2B) and the relative number of myotubes was 56.0% lower (*p* ≤ 0.05; Figure 2C) in hypoxia than in normoxia.

### 3.3. Development of a Novel Myoblast Fusion Assay to Analyze Interplay of Normoxic- and Hypoxic-Conditioned Myoblasts

After muscle injury, myoblasts are exposed to different oxygen partial pressures depending on the extent of vasculature damage. To investigate the fusion of normoxic- and hypoxic-conditioned myoblasts, we generated C2C12 cell lines stably expressing a green (GFP, used for normoxic C2C12 myoblasts) or red (RFP, used for hypoxic-preconditioned C2C12 myoblasts) fluorescence protein using the sleeping beauty transposon system. After a short selection period in normoxia, one-third of the cells with the highest fluorescence were sorted by FACS (Figure 3A,A’). Afterwards, C2C12*^GFP^* cells were exclusively cultured in 21% O_2_, while C2C12*^RFP^* cells were grown in 2% O_2_. A mixture of normoxic-conditioned GFP-positive and hypoxic-conditioned RFP-positive myoblasts was then used for differentiation assays. In these assays, the green, yellow, and red colors indicate the extent to which GFP- or RFP-positive myoblasts fused into myotubes (Figure 3B). Myoblasts cultured in 21% O_2_ were labeled with ‘N’ (normoxia) and those cultured in 2% O_2_ with ‘H’ (hypoxia). While the first letter denotes normoxic or hypoxic conditioning of C2C12 myoblasts, the letter after the arrow reflects the oxygen treatment applied during subsequent differentiation. For example, N&H▶Diff-H refers to a mixture of normoxic- (N) and hypoxic-conditioned (H) myoblasts that were subsequently differentiated in 2% O_2_ (Figure 3B).

To exclude a potential loss of myogenic differentiation ability of C2C12 cells due to stable insertion of the respective fluorophore coding sequence into the genomic DNA, we repeated the fusion experiments and found no significant differences for C2C12*^GFP^* in 21% (Figure 3C,C’) or C2C12*^RFP^* in 2% O_2_ (Figure 3D,D’) compared with non-transfected C2C12 cells (Figure 2A). Moreover, fusion of myoblasts leads to an obvious accumulation of fluorescence intensity (Figure 3C,D, arrowheads).

### 3.4. Abrupt Change of Oxygen Concentration Differentially Affects Myogenic Differentiation of Normoxic- and Hypoxic-Preconditioned Myoblasts

To analyze how the switch from normoxia to hypoxia or vice versa affects myogenic differentiation, we differentiated long-term, hypoxic-conditioned C2C12*^RFP^* myoblasts in normoxia (H▶Diff-N) and normoxic-conditioned C2C12*^GFP^* cells in 2% hypoxia (N▶Diff-H). Interestingly, when we directly compared H▶Diff-N (Figure 4A,A’) and N▶Diff-H (Figure 4C,C’) myotubes, we could not find significant differences in overall myogenic efficiency (Figure 4E), number of myotubes (Figure 4F), or size of myotubes (Figure 4G).

However, when we compared hypoxic-preconditioned cells differentiated in hypoxia (H▶Diff-H) (Figure 3D,D’) or normoxia (H▶Diff-N) (Figure 4A,A’), we detected a 5% and 19% increase in the efficiency of myogenic differentiation and the number of myotubes, respectively, and a 13% larger myotubes size in the H▶Diff-H group compared with H▶Diff-N myotubes (Figure 5).

In contrast, differentiation of normoxic myoblasts in 2% hypoxia (N▶Diff-H) reduced myogenic differentiation (Figure 4C,C’) compared with N▶Diff-N (Figure 3C,C’). Specifically, the total myotube area was 67% lower, there were 48% fewer myotubes, and they were 33% smaller compared with the N▶Diff-N myotubes group (Figure 5).

Collectively, our data suggest that (1) the negative effect of hypoxia on myogenic differentiation is reversible to some extent and (2) hypoxia leads to less but significant larger myotubes and is therewith partially involved in the hypertrophy of myotubes.

### 3.5. Fusion of Hypoxic-Conditioned Myoblasts with Normoxic-Conditioned Myoblasts Results in Larger Myotubes

Next, we investigated how a mixture of hypoxic- and normoxic-conditioned myoblasts under either normoxia or hypoxia affects myogenic differentiation. We found that a mixed population of myoblasts formed fewer myotubes when differentiated under hypoxia (H&N▶Diff-H, Figure 4D,D’) compared with normoxia (N&H▶Diff-N, Figure 4B,B’). Interestingly, mixed myoblasts differentiated in normoxia showed a 187% higher differentiation capacity (N&H▶Diff-N, *p* ≤ 0.0001, Figure 4E) when compared with myoblasts that were differentiated in hypoxia (N&H▶Diff-H, Figure 4E). In addition, there were 73% less N&H▶Diff-H myotubes and 162% less H&N▶Diff-H myotubes compared with myoblasts conditioned in hypoxia and differentiated under normoxia (H▶Diff-N, *p* ≤ 0.001, Figure 4F). However, when we differentiated a mixture of hypoxic- and normoxic-conditioned myoblasts under normoxia (N&H▶Diff-N), we found 239% or 285% larger myotubes (*p* ≤ 0.01) compared with either hypoxic-conditioned myoblasts differentiated under normoxia (H▶Diff-N) or normoxic myoblasts differentiated under hypoxia, respectively (N▶Diff-H, Figure 4G).

We made several additional observations. First, mixing hypoxic-preconditioned cells with normoxic cells inhibited myoblast fusion under normoxic differentiation (N&H▶Diff-N), as judged by the 42% reduction in total MYHE^+^ area (Figure 5). Second, there were 64% fewer myotubes compared with normoxic-precultured myoblasts differentiated under normoxia (N▶Diff-N). Third, mixed myoblasts were 14% larger when differentiated under hypoxia (H&N▶Diff-H) and 91% larger when differentiated under normoxia (N&H▶Diff-N) compared with normoxic myoblasts differentiated in normoxia (N▶Diff-N) (Figure 5). Finally, the mixture of long-term, hypoxic-conditioned and normoxic myoblasts (N&H▶Diff-N and H&N▶Diff-H) fused the least, resulting in the lowest number of myotubes (Figure 5).

### 3.6. Oxygen Tension Differentially Influences Cell Fusion among Different Precultured Cell Populations

To better understand the interaction between long-term, hypoxic-preconditioned and normoxic myoblasts during myogenic differentiation in hypoxia or normoxia, respectively, we divided the fused myotubes (MT) into three subgroups depending on which myoblasts were involved in the fusion process: MT^GFP^, MT^RFP^, and MT^GFP/RFP^. In addition to quantifying absolute values (Figure 6D–F), we calculated the relative abundance of MT^GFP^, MT^RFP^, and MT^GFP/RFP^ within each experimental group (Figure 6A–C), as the oxygen concentration during differentiation might differentially affect the fusion properties of subpopulations.

Determination of the relative area of individual myotube populations in mixed cultures revealed no differences between normoxic (N&H▶Diff-N) and hypoxic (H&N▶Diff-H) differentiation (Figure 6A). However, within the absolute area, each myotube population showed a significant increase in normoxic differentiation (N&H▶Diff-N) compared with differentiation in hypoxia (H&N▶Diff-H), reflected by a 128% minimum increase of the absolute MYH1E^+^ area in MT^GFP^ and a 316% maximum increase of the absolute MYH1E^+^ area in MT^GFP/RFP^ (Figure 6D).

Intriguingly, quantification of the relative number of myotube types showed a significant difference in myoblast fusion under normoxic (N&H▶Diff-N) and hypoxic conditions (H&N▶Diff-H) (Figure 6B), as indicated by the relative increase of MT^GFP/RFP^ and relative decrease of MT^GFP^ and MT^RFP^ myotubes compared with myoblast fusion of mixed cultures in 21% O_2_ (N&H▶Diff-N). However, the determination of absolute myotube number indicated that double-fluorescent myotubes did not differ between the differentiation in normoxia (N&H▶Diff-N) and hypoxia (H&N▶Diff-H), although the absolute numbers of MT^GFP^ and MT^RFP^ myotubes were significant lower in the H&N▶Diff-H group compared with the N&H▶Diff-N group (Figure 6E).

Moreover, the relative and absolute myotube size of MT^GFP/RFP^ was approximately 70% smaller in hypoxic differentiation (H&N▶Diff-H) compared with normoxic differentiation (N&H▶Diff-N) of mixed cultures (Figure 6C,F).

### 3.7. Hypoxia Leads to Delayed Expression of Genes Encoding Regulators of Myogenic Differentiation

To gain insight into the effect of oxygen on the molecular regulation of myoblasts and their differentiation into myotubes, we performed total RNA-sequencing of myotubes derived from a mixture of normoxic- and hypoxic-conditioned myoblasts that were subsequently differentiated under normoxia and hypoxia (N&H▶Diff-N and H&N▶Diff-H). RNA-Seq was performed at 24, 48, 72, 96, and 144 h after the onset of differentiation. To evaluate potential clustering, we performed a principal component analysis (PCA), which showed the clear separation of a time-dependent first principal component (PC1, Figure 7A, colored spheres) and a second, group-dependent principal component (PC2, Figure 7A, shapes). Moreover, the overall transcriptional landscape of the hypoxic-differentiated mixed C2C12 cells (H&N▶Diff-H, Figure 7A dots) is slightly delayed compared with the N&H▶Diff-N group (Figure 7A, triangles). While the 100 genes that contributed most to PC1 were myogenic-related marker genes (Appendix A), PC2 revealed segregation of genes associated with adaptation to hypoxia (Appendix A).

Differential gene expression (DEG) analysis revealed that 177 out of 6740 genes showed significantly different regulation between H&N▶Diff-H and N&H▶Diff-N groups at all time points (Figure 7B). All significant differentially expressed genes are shown in Appendix A. In addition, gene set enrichment analysis (GSEA) demonstrated that the three gene set hallmarks ‘hypoxia’, ‘glycolysis’, and ‘myogenesis’ were regulated at all time points (Figure 7C). All significant regulated gene set hallmarks are shown in Appendix A. Interestingly, at 72 and 96 h after the initiation of differentiation, a significant decrease in the ‘myogenesis’ gene set further highlighted an adaptive phase for the H&N▶Diff-H group compared with the normoxic mixed myoblast culture (N&H▶Diff-N, Figure 7C, lower panel).

To further investigate which genes are associated with differences of myogenic fusion at different oxygen concentrations of mixed myoblast cultures, we performed a hierarchical cluster analysis (HCA) for the top 50 genes out of the 177 continuously differentially regulated genes (Figure 7D). Intriguingly, HCA showed a clear separation of essential (increased expression over time) or inhibitory (decreased expression over time) myogenic genes in a temporal manner (Figure 7D). The nine most promising genes in this separation are as follows: *Adm* (adrenomedullin), encoding a vasodilator peptide hormone [15]; *Apln* (Apelin), known for the regulation of cell proliferation in smooth muscle cells [16]; *Car9* (Carboxy anhydrase 9), one of the most highly expressed genes in the hypoxic environment of solid tumors [17]; *Grp35* (G-protein related receptor 35), encoding an orphan receptor involved in ERK1/2 activation [18]; *Scara5* (Scavenger receptor class A member 5), encoding a ferritin receptor necessary for cell growth [19]; *Ptgfr* (prostaglandin F receptor), encoding a G-protein-coupled receptor involved in prostaglandin signaling; *Ankrd1* (Ankyrin repeat domain 1), mainly expressed in type 1 skeletal muscle fibers [20]; *Ldb3* (LIM domain binding 3), encoding for Z-band alternatively spliced PDZ-motif protein [21]; *Pdlim3* (PDZ And LIM Domain 3), involved in p38 MAPK activation (Figure 7E) [22].

## 4. Discussion

The main findings of this study are that chronic exposure to hypoxia or normoxia has a differential effect on myoblast differentiation and that a mixture of normoxic- and hypoxic-conditioned myoblasts forms larger myotubes than when myoblasts are conditioned in normoxia or hypoxia alone. Additionally, we identified genes whose expression changes significantly in response to hypoxia and/or during differentiation.

### 4.1. Long-Term Hypoxia Has No Significant Impact on Myoblast Morphology or Proliferation but Leads to Reduced Myogenesis

During somitogenesis, a community effect of 30–40 mesodermal cells is required so that these cells differentiate into myoblasts [23]. Studies have shown that myogenesis is dependent on the expression cell–cell adhesion molecules as N-cadherin (CDH2), which promote the stable expression of myogenic factors [24] and the accumulation of p21 and p27 cyclin-dependent kinase inhibitors, involved in cell cycle withdrawal [25,26,27]. As several cadherins, including CDH2, were differentially regulated between N&H▶Diff-N and H&N▶Diff-H from 24 h upwards (Appendix A), we suggest that the changes in the cell aspect ratio as well as the reduced myoblast fusion under hypoxic conditions are, at least partially, dependent on the expression of hypoxia-sensitive cadherin genes.

Furthermore, while short-term hypoxia can induce an increased proliferation of mouse satellite cells [28] or human primary myoblasts [29] at 2% O_2_ in vitro, long-term hypoxic culture of human primary myoblast led to an adjustment of proliferation between normoxic and hypoxic conditions [29], which is consistent with our data showing no significantly different proliferation of C2C12 cells between 21% and 2% O_2_.

As previously described by Di Carlo et al., severe acute hypoxia (<1% O_2_) leads to accelerated degradation of MYOD and thereby inhibits differentiation of C2C12 myoblasts [30]. Here, we demonstrated that an oxygen level of 2% O_2_ over a period of 4 days likewise results in reduced myotube formation, indicating a very quick adjustment to hypoxia. Interestingly, our data showed that not only hypoxic differentiation of hypoxic-precultured myoblasts (H▶Diff-H) led to a reduction in myotube size and number of 40–60%, but we also showed that this negative effect of hypoxia is largely irreversible in differentiation in normoxia (H▶Diff-N).

Myoblast fusion is highly dependent on available energy sources such as ATP [31] and ATP synthesis is impaired in hypoxia [32]. Compared with myoblasts, which are mainly dependent on glycolytic energy production [33], myotubes are less able to maintain ATP levels and turnover, and thus, die within hours [34]. Therefore, the reduction of myogenesis under hypoxic conditions is a multimodal process consisting of at least reduced fusion and subsequent cell death due to energy restrictions.

### 4.2. Hypoxia Leads to Increased Synergy between Hypoxia- and Normoxia-Cultured Myoblasts

In the case of acute ischemic events, it has been established that regeneration can only occur if blood flow to the affected area is restored [11] and stemlike satellite cells can migrate into (re-)oxygenated tissue [35]. Interaction of myoblasts/satellite cells also occurs after intramuscular transplantation of satellite cells, as shown by a potential novel therapy approach for ischemic muscle injuries [36,37,38]. Thus, our study mainly focused on the interaction of normoxic and hypoxic precultured myoblasts during the fusion process in normoxic and hypoxic environments.

Intriguingly, both mixed myoblast groups (N&H▶Diff-N and H&N▶Diff-H), regardless of their environmental oxygen exposure, showed a smaller total myotube area and number of myotubes, and a remarkably greater relative myotube size compared with standardized N▶Diff-N group (Figure 5). This suggests that the leading mechanism in the mixed myoblast fusion process is hypertrophy, which results in fewer but larger fused myotubes in an oxygen-dependent manner. Moreover, this hypertrophy seems to depend on the direct fusion of normoxic- and hypoxic-preconditioned myoblasts, suggesting a synergistic effect.

### 4.3. Hypoxic Transcriptional Changes Are Partially Reversible during Myogenic Differentiation

The potential of myoblasts to fuse was not completely lost within the hypoxic environment, as MTORC1 signaling involved in myoblast proliferation and differentiation [39] was still expressed in the H&N▶Diff-H group.

During prolonged hypoxia, locally released proinflammatory cytokines induce an anabolic response in skeletal muscle cells, which ultimately results in muscle damage and atrophy [40]. However, short hypoxic periods of inflammation are also crucial for adaptive remodeling of skeletal muscle tissue. For example, prostaglandins (PGs), members of inflammatory mediators, have been implicated in multiple stages of myogenesis in vitro, including proliferation, differentiation [41], myoblast survival [42] and fusion [43], and prostaglandin 2α (PGF2α) leads to muscle hypertrophy via the PI3K/ERK/mTOR signaling pathway in C2C12 myotubes [44]. Interestingly, our new data identified *Ptges3* (Prostaglandin E Synthase 3) and *Ptges3l* (Prostaglandin E Synthase 3 Like) as novel candidates that are involved in oxygen-dependent hypertrophy.

In addition, we also identified hypoxia-dependent regulation of *Apln* within the mixed hypoxic group (H&N▶Diff-H). However, *Apln* has exclusively been associated with hypoxia in cardiomyocytes [45] or age-related muscle wasting (sarcopenia) in rodents and humans [46]. Thus, it can be hypothesized that the regulation of APLN is, at least partially, oxygen-driven, as the oxidative capacity of skeletal muscles is reduced during aging [47] and APLN is associated with energy metabolism [48].

Compared with myogenic differentiation of mixed myoblasts in hypoxia, myoblasts/myotubes of the N&H▶Diff-N group showed a significantly faster and higher upregulation of LIM domains, which are known to be involved in myogenesis [49,50]. Intrudingly, we identified *Pdlim3* and *Ldb3* as oxygen-sensitive genes that are associated with hypertrophic cardiomyopathies [51,52].

In vivo, transcriptional changes of whole muscles exposed to moderate hypoxia (11.2%) have been reported after 48 h [53], which is consistent with our results showing a transcriptional change between 24 and 72 h of myogenic differentiation in hypoxia. However, as the transcriptional response to hypoxia is relatively fast in vitro, further time points should be evaluated in future experiments. Moreover, we observed a partial delayed myogenic differentiation of the H&N▶Diff-H within a period of 6 days, assuming that differentiation in hypoxia leads to an initial “hypoxic shock”, which seems to be partially reversible. Thus, the investigation of longer observation periods would be necessary.

### 4.4. Within Hypoxia, Skeletal Muscle Hypertrophy Shows Parallels to Cancer Cell Behavior

Recent studies have revealed increased glycolysis and metabolic reprogramming of skeletal muscle cells during hypertrophy [54], which are similar to those in cancer cells [55]. Our results sustain those findings, as upregulation of several cancer-associated genes was found within mixed myoblast cultured in a hypoxic environment. For example, *Car9*, a well-studied gene in solid tumors such as renal clear cell carcinoma [56], which is mainly associated with tumor-related hypoxia, was upregulated in H&N▶Diff-H over all time points. *Car9* has an upstream hypoxia-responsive element (HRE) binding site [56] that appears to be epigenetically active in zebrafish muscles in response to hypoxia [57].

Although no association with skeletal muscle cells has been reported to date for the KRAS-pathway, its upregulation was observed in the mixed hypoxic myoblast group (H&N▶Diff-H). The KRAS-pathway is known to be involved in cellular proliferation associated with colon, pancreatic, and lung cancer [58] and has previously been shown to have an influence on HIF1A translational regulation within hypoxia [59]. Moreover, the *Scara5* gene, which is associated with tumor suppression in liver [60], lung [61], and breast cancer [62], and osteosarcoma [63], showed upregulation in the H&N▶Diff-H group. It has been reported to have a negative correlation with VEGFA, one of the major target genes of HIF1A, leading to downregulation of angiogenesis. Further investigations revealed a SCARA5-dependent decrease of AKT and ERK phosphorylation [62], which are both also involved in hypoxic-adjustment of skeletal muscle cells [1].

Remarkably, many of the genes found in our study interact with the PI3K/ART/mTOR pathway, for example, SCARA5 and GPR35 (G-coupled receptor 35) [18,62]. In addition, muscle hypertrophy and cancer cells show increased IGF-AKT1-mTORC1 or reduced myostatin signaling [55]. Thus, the mTOR pathway may also play a major role in hypoxic-adjustment of skeletal muscle cells.

## 5. Conclusions

Taken together, the interaction of hypoxic and normoxic myoblasts leads to a decrease in the total number of myotubes but is likely to result in more hypertrophic myotubes when fused together. This hypertrophic process is possibly based in part on cance-like metabolic reprogramming strategies during hypoxic myogenesis. However, additional loss- and gain-of-function experiments with the promising candidate genes are necessary to reveal further confirmations.

## Figures and Tables

**Figure 1 cells-11-01059-f001:**
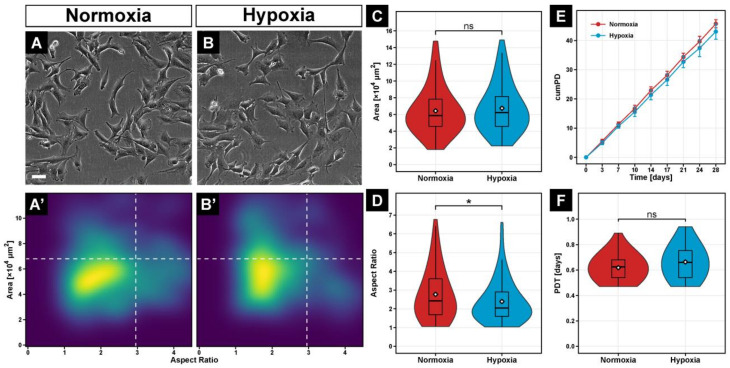
Phase-contrast images of C2C12 cells grown continuously for 28 days at 21% O_2_ (**A**) or 2% O_2_ (**B**). Quantification of cell area and aspect ratio illustrates the morphology distribution between normoxic (**A’**) and hypoxic (**B’**) cultured cells. Statistical analysis of individual cell morphology parameters revealed a significant decrease in aspect ratio (**D**) but not area (**C**). The effect of long-term hypoxia on cell proliferation, evaluated by cumulative population doublings (cumPD) (**E**) and population doubling time (PDT) (**F**) over a period of 28 days showed no significant differences between the two oxygen levels. Box plots represent the median, quartiles, range, and mean (white diamond). Scale bar: 100 µm. Significance level *: *p* ≤ 0.05, ns—not significant.

**Figure 2 cells-11-01059-f002:**
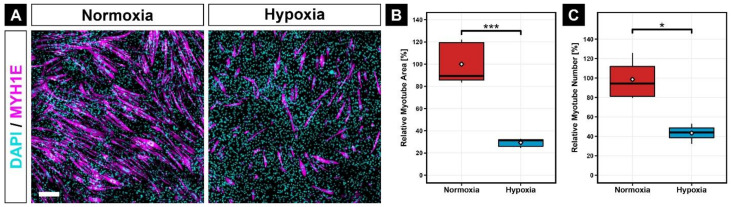
Immunocytochemistry against MYH1E (magenta) after 7 days of myogenic differentiation at 2% O_2_ showed a great reduction of myotube formation compared to 21% O_2_ (**A**). Quantification of relative myotube area (**B**) and relative myotube number (**C**) revealed significantly lower values in hypoxia than in normoxia. Scale bar: 200 µm. Box plots represent the median, quartiles, range and mean (white diamond). Significance levels: * equals *p* ≤ 0.05; *** equals *p* ≤ 0.001.

**Figure 3 cells-11-01059-f003:**
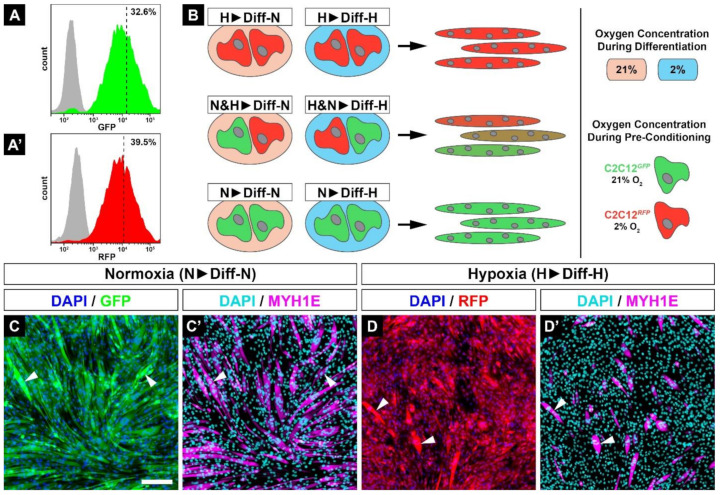
C2C12 cells were stably transfected with fluorescence proteins, sorted for high GFP (**A**) or RFP (**A’**) fluorescence intensity by FACS, and afterwards continuously cultured at 21% (C2C12*^GFP^*) or 2% O_2_ (C2C12*^RFP^*). Myogenic differentiation assays were performed with single or mixed colored cell lines. Thus, coloring depends on the fusion of various myoblasts populations, resulting in green (C2C12*^GFP^*), red (C2C12*^RFP^*), or dual-colored (C2C12*^GFP/RFP^*) myotubes (**B**). Immunocytochemistry against MYH1E shows that the newly generated reporter C2C12 cell lines kept their fusion ability (**C’**,**D’**) and C2C12*^RFP^* cells showed fusion deficiency in hypoxia (**D**,**D’**) when compared with C2C12*^GFP^* (**C**,**C’**). Scale bar: 200 µm.

**Figure 4 cells-11-01059-f004:**
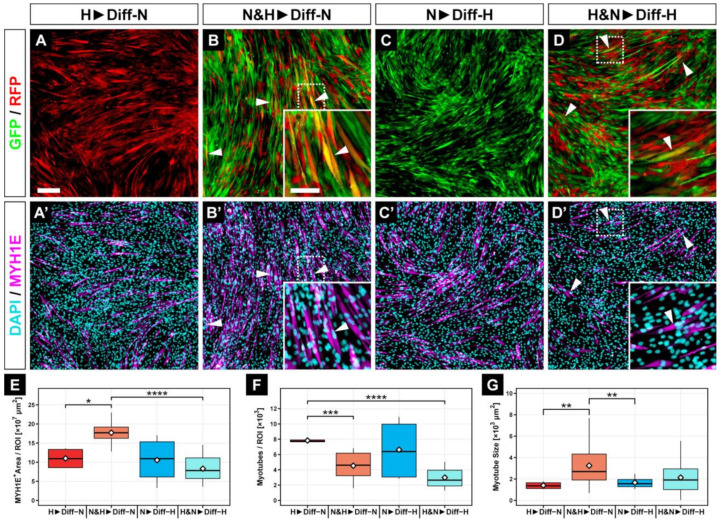
Representative fluorescence microscope images for GFP/RFP (**A**–**D**) and for MYH1E and DAPI (**A’**–**D’**). C2C12*^RFP^*, hypoxic-conditioned cells transferred to 21% O_2_ (H▶Diff-N, **A**,**A’**) or mixed C2C12 cultures in 2% O_2_ (H&N▶Diff-H, **D**,**D’**) showed less myoblast fusion compared with normoxic cells differentiated in hypoxia (N▶Diff-H, **C**,**C’**) or mixed C2C12 cultures differentiated in normoxia (N&H▶Diff-N, **B**,**B’**). Moreover, C2C12*^GFP^* and C2C12*^RFP^* cells were able to fuse in mixed culture settings (**B**,**D**, arrowheads). While the mixed group had the largest MYH1E^+^ area (**E**) and the greatest myotube size at 21% O_2_ (**G**), the myotube number did not increase with mixing of the two cell populations (**F**). Scale bars: 200 µm. Bar plots represent the median, quartiles, range, and mean (white diamond). * *p* ≤ 0.05. ** *p* ≤ 0.01. *** *p* ≤ 0.001. **** *p* ≤ 0.0001.

**Figure 5 cells-11-01059-f005:**
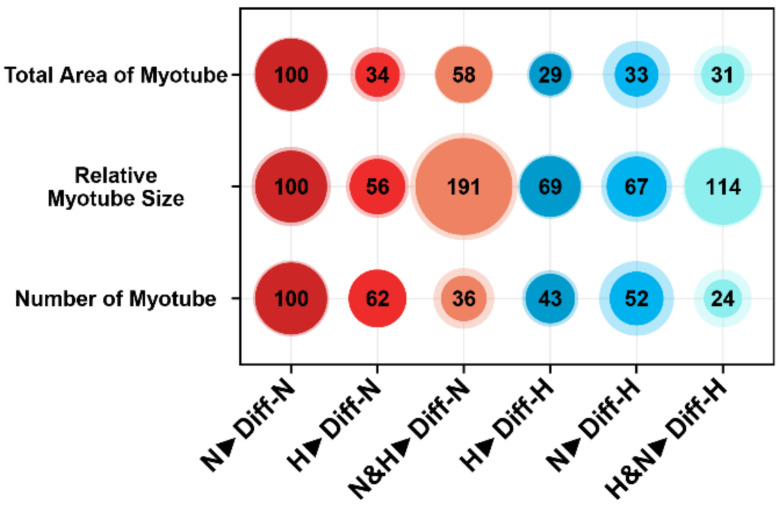
Bubble plots of all investigated groups normalized to N▶Diff-N. Mixed cell cultures showed the greatest relative myotube size, independent of their oxygen environment, even though the number of myotubes was decreased compared with the other groups. Despite low number of myotubes, the total area was greater within N&H▶Diff-N compared with all other groups.

**Figure 6 cells-11-01059-f006:**
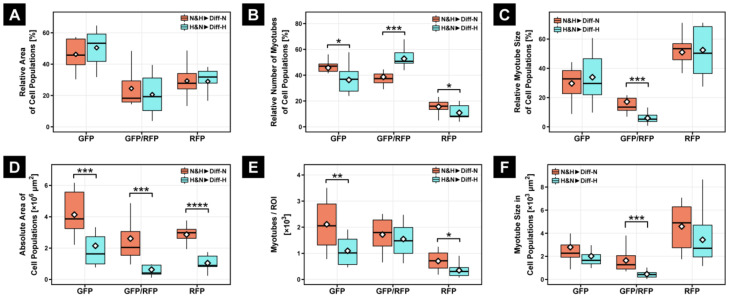
Within the subpopulation, there was no significant difference in relative area of cell population (**A**); however, quantification of the absolute area of all subpopulations at 21% O_2_ (N&H▶Diff-N) showed a significant increase compared with H&N▶Diff-H (**D**). While the relative number of mixed myotubes under hypoxic differentiation condition (H&N▶Diff-H) revealed a significant increase in double-fluorescent myotubes compared with differentiation in normoxia (N&H▶Diff-N; **B**), the absolute number of myotubes did not show similar results (**E**). The size of myotubes at 21% O_2_ was significantly greater within mixed myotubes, regardless of absolute or relative view, when compared with the H&N▶Diff-H group (**C**,**F**). Bar plots represent the median, quartiles, range, and mean (white diamond). * *p* ≤ 0.05. ** *p* ≤ 0.01. *** *p* ≤ 0.001. **** *p* ≤ 0.0001.

**Figure 7 cells-11-01059-f007:**
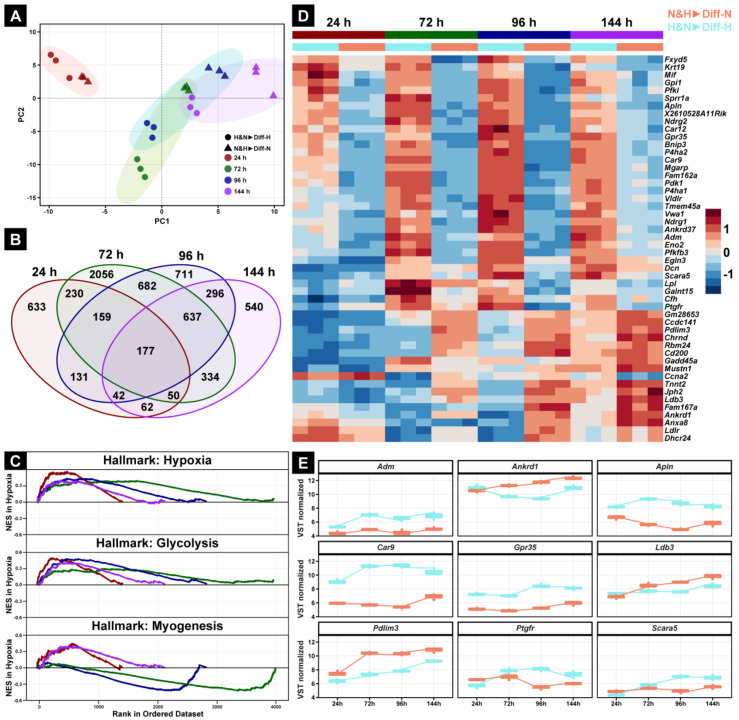
Comparison of gene expression between N&H▶Diff-N and H&N▶Diff-H. The first principal component (PC1) and second principal component (PC2) cluster analysis showed no initial difference between N&H▶Diff-N and H&N▶Diff-H 24 h after the onset of myogenic differentiation (**A**). The Venn diagram shows 177 genes present at all analyzed time points (**B**). Visualization of hypoxic-dependent hallmarks over 144 h shows the most positively (NES > 0) or negatively (NES < 0) regulated gene sets in the myoblast group H&N▶Diff-H compared with the normoxic group, N&H▶Diff-N (**C**). The heat map (**D**) includes the top 50 genes with the greatest variance within 144 h out of the 177 genes shown in (**B**). A positive value indicates higher expression; a negative value indicates lower expression. The nine most promising genes were extrapolated from the heat map and their activation under both oxygen conditions was assessed and directly compared over 144 h. (**E**) Genes are sorted alphabetically. NES—normalized enrichment score.

## Data Availability

Gene expression data are deposited at https://www.ncbi.nlm.nih.gov/geo. All in vitro data are available from the corresponding author upon reasonable request.

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
