# Peer review of "Fusion of Normoxic- and Hypoxic-Preconditioned Myoblasts Leads to Increased Hypertrophy"

_cells, 2022, doi:10.3390/cells11061059_

Round 1
Reviewer 1 Report
The paper describes the differences between C2C12 cell fusion in different culture conditions, showing the influence of oxygen concentration on the process. It also shows that cell history influences its ability to fuse into myotubes and that myotubes differ depending on the past of cells that have formed them.
The strongest element of the paper is elegant assay authors constructed to study fusion of two classes of cells with different history.
The manuscript is well written and the presented material is new, interesting and important.
It would be however good if authors could answer the following questions:
Why are cells presented in fig 1B clearly less numerous than in fig 1A if there is no difference in proliferation rate between hypoxia and normoxia?
Cells grown in hypoxia look also more motile or at least have more active lamellipodia than those in normoxia. The question arises if they different in motile behavior? If yes, do they adhere in the same way to the substratum? If the rate of adhesion is different in normoxic and hypoxic cells, maybe this is the reason why the rate of fusion between hypoxic and normoxic than between hypoxic and hypoxic or normoxic and normoxic. The moment of fusion for each class may be shifted in time due to the rate of adhesion.
Minor issue:
The great oxidation event took place about 2.4 billion, not million years ago.
Reviewer 2 Report
This study of Pircher et al, titled “Fusion of normoxic and hypoxic pre conditioned myoblasts leads to increased hypertrophy”, covers an upcoming and important field, with fertile ground for development and advancement. The authors provide an adequate, well written and informative introduction to the article, giving a proper background to the investigation, hypoxia as a physiological and pathological event in muscle function, focused literature and existing data. The subsequent structure of the paper, i.e. Materials and methods, Results, and Discussion are also written well. Authors presented interesting data and proved that prolonged hypoxy negatively impacts on myoblast fusion. Moreover fusion/interaction of C2C12 myoblasts cultured in normoxia and hypoxia leads to decreased number of myotubes with hypertrophic morphology.
However, I have a few comments:
- In materials and methods section in Myoblast fusion assay Authors cultured 450 000 cells per well of 6-well plates; i.e. per 10 cm2 – this way to much or this is a mistake.
- In in vitro culture there is no possibility to obtain myofibers thus Authors should change it to myotubes.
Minor corrections: Spelling errors throughout the manuscript should be revised.
In short, this paper should accepted after minor correction.
Round 2
Reviewer 1 Report
Now everything looks good.
Reviewer 2 Report
I'm satisfied with the Authors' responses. I recommend acceptance of the manuscript and its publication in its present form.